# Ultrafast acousto-optic mode conversion in optically birefringent ferroelectrics

Mariusz Lejman[1], Gwenaelle Vaudel[1], Ingrid C. Infante[2], Ievgeniia Chaban[1], Thomas Pezeril[1], Mathieu Edely[1], Guillaume F. Nataf[3,4], Mael Guennou[3], Jens Kreisel[3,5], Vitalyi E. Gusev[6], Brahim Dkhil[2] & Pascal Ruello[1]

The ability to generate efficient giga–terahertz coherent acoustic phonons with femtosecond laser makes acousto-optics a promising candidate for ultrafast light processing, which faces electronic device limits intrinsic to complementary metal oxide semiconductor technology. Modern acousto-optic devices, including optical mode conversion process between ordinary and extraordinary light waves (and vice versa), remain limited to the megahertz range. Here, using coherent acoustic waves generated at tens of gigahertz frequency by a femtosecond laser pulse, we reveal the mode conversion process and show its efficiency in ferroelectric materials such as $BiFeO_3$ and $LiNbO_3$. Further to the experimental evidence, we provide a complete theoretical support to this all-optical ultrafast mechanism mediated by acousto-optic interaction. By allowing the manipulation of light polarization with gigahertz coherent acoustic phonons, our results provide a novel route for the development of next-generation photonic-based devices and highlight new capabilities in using ferroelectrics in modern photonics.

[1] Institut des Molécules et Matériaux du Mans, UMR CNRS 6283, Université du Maine, Av O. Messiaen, 72085 Le Mans, France. [2] Laboratoire Structures, Propriétés et Modélisation des Solides, CentraleSupélec, UMR CNRS 8580, Université Paris-Saclay, 92295 Châtenay-Malabry, France. [3] Materials Research and Technology Department, Luxembourg Institute of Science and Technology, 41 Rue du Brill, L-4422 Belvaux, Luxembourg. [4] SPEC, CEA, CNRS, Université Paris-Saclay, CEA Saclay, 91191 Gif sur Yvette, France. [5] Physics and Materials Science Research Unit, University of Luxembourg, 41 Rue du Brill, L-4422 Belvaux, Luxembourg. [6] Laboratoire d'Acoustique de l'Université du Maine, UMR CNRS 6613, Université du Maine, 72085 Le Mans, France. Correspondence and requests for materials should be addressed to P.R. (email: pascal.ruello@univ-lemans.fr) or to V.E.G. (email: vitali.goussev@univ-lemans.fr).

Research towards shaping the light spectrum (its amplitude, polarization and frequency)[1–3], including recent supersymmetry optics in synthetic Bragg filters offering new ways of mode-conversion and spatial multiplexing[4,5], herald promising perspectives in modern photonics for optical imaging and communication applications. Moreover, because of a ceaseless demand of new and tunable processes for light modulation at high rate[6,7], novel routes with efficient ultrafast mechanisms are highly desired. Fostered by a wide range of applications, devices using stimulated birefringence (electro-optics and acousto-optics) through solid state anisotropic media are already commercially available for light modulation. For instance, acousto-optic technology[8] is well-established with acousto-optic modulators and shifters, dispersive filters[9] or polarization converters as sketched in Fig. 1a (ref. 10). However, today devices use sound waves at MHz (radiofrequency) range, while the next generation of signal processing and communications will require a new breakthrough into GHz frequencies. Therefore, ultrafast processes, where light is interacting with acoustic phonons, are currently under active considerations as potential solutions for future devices. Consequently, such mechanisms are at the heart of several emerging research fields at the frontier between photonics and phononics that includes coupled optical/mechanical cavities[11], optomechanics[12] or optomechanical crystals[13]. Interestingly, fast progress in the generation and detection of GHz–THz coherent acoustic phonons permits today to envision their use for ultrafast scattering of photons. Ultrafast coherent acoustic waves are indeed able to modulate the mass density, that is, to change the refractive indices (photoelastic effect) or induce displacements of surfaces/interfaces.

Here, on the way towards ultrafast acousto-optic devices, we report the observation of ultrafast light mode conversion induced by coherent acoustic phonons through uniaxial birefringent media, namely the ferroelectrics $BiFeO_3$ (BFO) and $LiNbO_3$ (LNO). Our analysis shows that this process is only detectable, thanks to specific photoelastic coefficients, which are much larger for BFO and LNO than for the canonical non-ferroelectric birefringent calcite ($CaCO_3$) medium. Moreover, based on estimates, we argue that the as-evidenced acousto-optic mode conversion at ultrafast time-scale is likely to take place within a few nanometre spatial scale, that is, at the sharp front of the coherent acoustic pulse induced by femtosecond laser excitation, allowing integration of the phenomenon into nanostructures such as thin films. Finally, our results pave the way towards all-optical ultrafast light modulation as illustrated for polarization shaping through a promising and potentially fast implemented concept sketched in Fig. 1b. Instead of encoding the optical information with an external electric field as done in classical acousto-optic devices (Fig. 1a), the idea here is to use alternatively a laser pulse as external stimulus and take advantage of the ultrafast acoustic waves generated (Fig. 1b). This all-optical mechanism further allows to deal with GHz–THz signals pushing up the frequencies used in the actual MHz devices. Knowing that LNO is already used in a wide spectrum of optical applications and is commercially available, such material can be rapidly integrated into the device concept we propose (Fig. 1b). Moreover, we also reveal that in addition to usual LNO material, smaller band gap ferroelectrics such as the prototypical BFO compound, which displays comparable photoelastic responses, hold promises as ferroelectric-based acousto-optic devices enriching their already wide photoferroelectric[14] as well as polaritonic[15,16] potentialities.

known as the Brillouin–Mandelstam scattering process and has been widely used to probe acoustic phonons dynamics in condensed matter including solids and liquids, thanks to Brillouin spectroscopy[17–20]. This scattering process is based on the local modulation of the refractive index of the matter caused by the acoustic phonons strain field. In turn, this creates inhomogeneities of the optical properties that induce light scattering. This is the so-called photoelastic, acousto-optic or elasto-optic effect. In the quantum approach, the Brillouin process can also be viewed as a collision between a photon and an acoustic phonon. In the case of uniaxial birefringent media, the Brillouin process becomes much richer, as photons do not propagate with the same velocity depending on their polarization, that is, ordinary (o) or extraordinary (e). Therefore, it is predicted that during the Brillouin process up to four Brillouin modes exist in anisotropic medium[21]. This is due to the light wave momentum conservation $\mathbf{k}_s - \mathbf{k}_i = \mathbf{q}_{ph}$ ($\mathbf{k}$ and $\mathbf{q}_{ph}$ are the momenta of photon and phonon respectively, s and i subscripts stand for scattered and incident probe light beams), which can be achieved following four scattering processes (o)-(o), (e)-(e), (o)-(e) and (e)-(o) as depicted in Fig. 1c for the case of uniaxial media (in normal incidence with the backscattering geometry, $\alpha = 0$, the vectors $\mathbf{q}_{ph}^{(o)-(e)}$ and $\mathbf{q}_{ph}^{(e)-(o)}$ are identical). Despite the theoretical expectations, only few experimental observations of these multiple Brillouin–Mandelstam processes have been reported in the literature and are limited to only non-coherent acoustic phonons (thermally excited acoustic phonons). The two modes corresponding to the scattering of ordinary and extraordinary modes, that is, (e)-(e) and (o)-(o) processes, have been often demonstrated in conventional Brillouin spectroscopy experiments[22–25], whereas the mode-converted Brillouin peaks, that is, (e)-(o) and (o)-(e) processes, have been reported only a few times[21].

Thanks to time-resolved optical measurements, the Brillouin process can be nowadays explored in the time domain and at short time scale following the picosecond acoustic interferometry scheme[26–28]. In this regime, the scattering process is no more achieved with thermal acoustic phonons, but with coherent acoustic phonons generated by a femtosecond laser pulse according to different possible generation mechanisms[26,29,30]. The coherent acoustic phonons are first generated in the material (on the surface as depicted in Fig. 1d) and a delayed probe light beam can detect the phonons propagation. The light probe–phonon scattering process can be monitored, thanks to an interferometric effect involving the femtosecond light probe pulse reflection at the free surface (beam 1 in Fig. 1d) and the light probe scattered by the moving coherent acoustic phonons front (beam 2 in Fig. 1d). Following a phase matching rule, this interference process gives rise in the time domain to a modulation of the detected light probe intensity with a period in time, which is the inverse of the detected acoustic mode frequency $f_B$ (so-called Brillouin oscillations). For the simple case of an isotropic medium and for probe light in the visible diapason, for which the elastic scattering approximation is precise enough to evaluate the frequency of the phonon, we obtain the following Brillouin frequency:

$$f_B \simeq 2V\sqrt{n^2 - sin^2(\psi)}/\lambda \qquad (1)$$

where $n$ is the refractive index, $V$ the sound velocity, $\lambda$ the probe wavelength in vacuum and $\psi$ the probe incidence angle[27]. $\alpha$ and $\psi$ are linked by the Snell–Descartes refraction law ($sin(\psi) = n\,sin(\alpha)$).

As in an optically anisotropic crystal up to four Brillouin processes are possible, up to four Brillouin frequencies could be detected with this time-domain experiment. In the particular case of our experimental configuration that corresponds to the

## Results

### Brillouin–Mandelstam process in optically anisotropic media.
The interaction of acoustic phonons with photons is historically

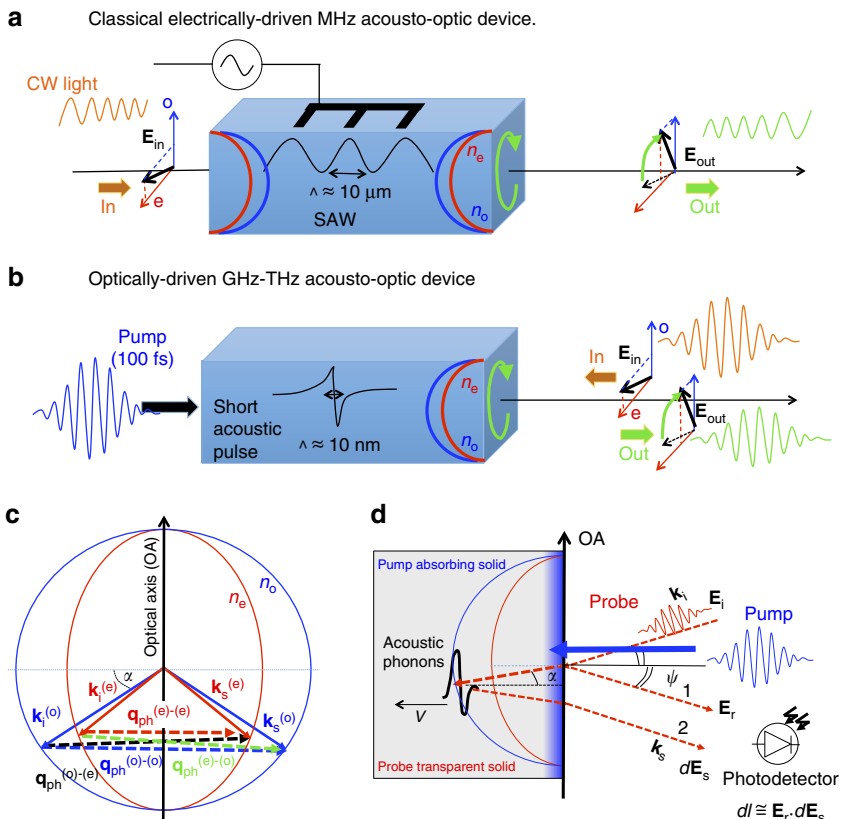

**Figure 1 | Acousto-optic devices principles and Brillouin–Mandelstam scattering processes in uniaxial birefringent crystal.** Principle of acousto-optic devices: (**a**) current electrically driven MHz collinear acousto-optic devices based on the modulation of continuous waves (cw) light polarization (light electric field $\mathbf{E}_{in}$ converted into $\mathbf{E}_{out}$) assisted by acoustic waves (example of SAW generated by piezoelectric materials usually[10]). The symbols (e) and (o) mean extraordinary and ordinary light components. (**b**) Proposed concept of light-driven GHz–THz acousto-optic devices based on high-frequency coherent acoustic phonons (generated with femtosecond laser pulses). In this latter scheme, a light mode conversion (light electric field $\mathbf{E}_{in}$ to $\mathbf{E}_{out}$) is induced by a very localized acoustic strain gradient (10 nm) preliminary photoinduced by a femtosecond laser. The acousto-optic interaction results in the backscattering geometry to a time-dependent modulation of the Brillouin light intensity controlled by the moving acoustic front (ps time-scale modulation). The intensity and polarization of the scattered light can be tailored by engineering the spectrum of the incident picosecond acoustic pulse. (**c**) Four possible photon–phonon scattering processes: the ordinary ((o)-(o)) and extraordinary ((e)-(e)) light scattering, as well as the mode conversion processes ((e)-(o) and (o)-(e)). OA is the optical axis. (**d**) Principle of a two-colour pump–probe time-resolved experiment: the pump light is absorbed in the near surface to generate acoustic phonons, while the probe that penetrates inside the solid is able to detect these propagating acoustic phonons ($V$ is the sound velocity). The reflected (beam 1) and scattered (beam 2) probe beams interfere in the photodetector ($\mathbf{E}_r d\mathbf{E}_s$) to give rise to Brillouin oscillations.

probing at normal incidence with the backscattering geometry ($\psi = \alpha = 0$), two scattering processes are degenerated in energy, that is, (e)-(o) and (o)-(e). Consequently, three different Brillouin frequencies are expected with:

$$f_{o-o} = 2Vn_o/\lambda \tag{2}$$

$$f_{o-e} = f_{e-o} = V(n_o + n_e)/\lambda \tag{3}$$

$$f_{e-e} = 2Vn_e/\lambda \tag{4}$$

where $n_o$ and $n_e$ are the ordinary and extraordinary refractive indices, respectively. The existence of these Brillouin processes is however completely determined by the efficiency of the acousto-optic process through the strength values of the photoelastic constants $p_{ij}$, as we will discuss later on. Despite numerous studies, only one work reports on the time-domain Brillouin scattering in anisotropic medium[31]. However, only the (e)-(e) and (o)-(o) processes were observed in the Brillouin spectrum, which prevents the manipulation of the light polarization at the picosecond time scale. In the following, we will show in contrast that the light mode-conversion ((o)-(e)) assisted by GHz coherent acoustic phonons becomes possible in ferroelectric

materials such as LNO and BFO, whereas this process remains not efficient enough in the canonical birefringent $CaCO_3$ (CCO) crystal. As we will see, this peculiar behaviour comes from exceptionally large $p_{41}$ coefficients that BFO and LNO ferroelectrics have and which are probably linked to the ferroelectric properties, although no direct relationship has been established yet.

**Experimental time-resolved Brillouin signals.** The preparation and characteristics of BFO, CCO and LNO samples, as well as the time-resolved optical experiments principle are described in Methods. Additional details regarding CCO and LNO samples can be found in Supplementary Figs 1–3 and Supplementary Note 1, whereas complements about ultrafast optics methods will be found in Supplementary Fig. 4 and Supplementary Note 2.

We discuss first now the results obtained with BFO, which has become recently a widely investigated material for its multiferroic and, more recently, interesting optical properties[32]. It also has the largest birefringence among the three solids under investigation (Table 1), which is a favourable situation to reveal the three Brillouin processes. In Fig. 2a, the geometry conditions used in

**Table 1 | Optical refractive indices ($n_o$ and $n_e$) and photoelastic coefficients $p_{ij}$ of birefringent crystals.**

| Materials | $CaCO_3$ | $LiNbO_3$ | $BiFeO_3$ |
|---|---|---|---|
| Band gap $E_g$ (eV) | 6 | 3.7 | 2.6–2.8 |
| Optical refractive indices | ref. 35 | refs 47,48 | ref. 44 |
| Ordinary index $n_o$ | 1.658 (590 nm) | 2.272 (590 nm) | 2.85 (800 nm) |
| | | 2.44 (400 nm) | 3.05 (590 nm) |
| Extraordinary index $n_e$ | 1.486 (590 nm) | 2.187 (590 nm) | 2.65 (800 nm) |
| | | 2.33 (400 nm) | 2.8 (590 nm) |
| Birefringence $\Delta n = n_e - n_o$ | − 0.172 (590 nm) | − 0.085 (590 nm) | − 0.2 (800 nm) |
| | | − 0.11 (400 nm) | − 0.25 (590 nm) |
| Photoelastic coefficients | ref. 22 | ref. 49 | |
| $p_{21}$ | 0.147 | 0.072 | - |
| $p_{31}$ | 0.241 | 0.178 | - |
| $p_{41}$ | − 0.036 | 0.155 | - |
| $p_{14}$ | − 0.011 | 0.070 | |
| $p_{51}$ | 0 | 0 | 0 |
| Brillouin modes relative amplitudes | | | |
| $A_{o-o}^{max}$ | ≈53% | ≈13% | ≈31% |
| $A_{e-e}^{max}$ | ≈47% | ≈58% | ≈50% |
| $A_{e-o}^{max}$ | ≈0 | ≈29% | ≈19% |

The values of the optical indices are given for the relevant optical wavelengths used in this work. The presented photoelastic coefficients are those that are relevant with the scattering geometry considered in our work. To evidence some correspondence between time-resolved Brillouin modes amplitudes and photoelastic coefficients, we give the relative (%) peak-to-peak amplitudes of squared cosinus function (Fig. 6) describing the experimental acousto-optic processes ($A_{o-o}^{max}$, $A_{e-e}^{max}$ and $A_{e-o}^{max}$ for ordinary, extraordinary and mode-conversion processes).

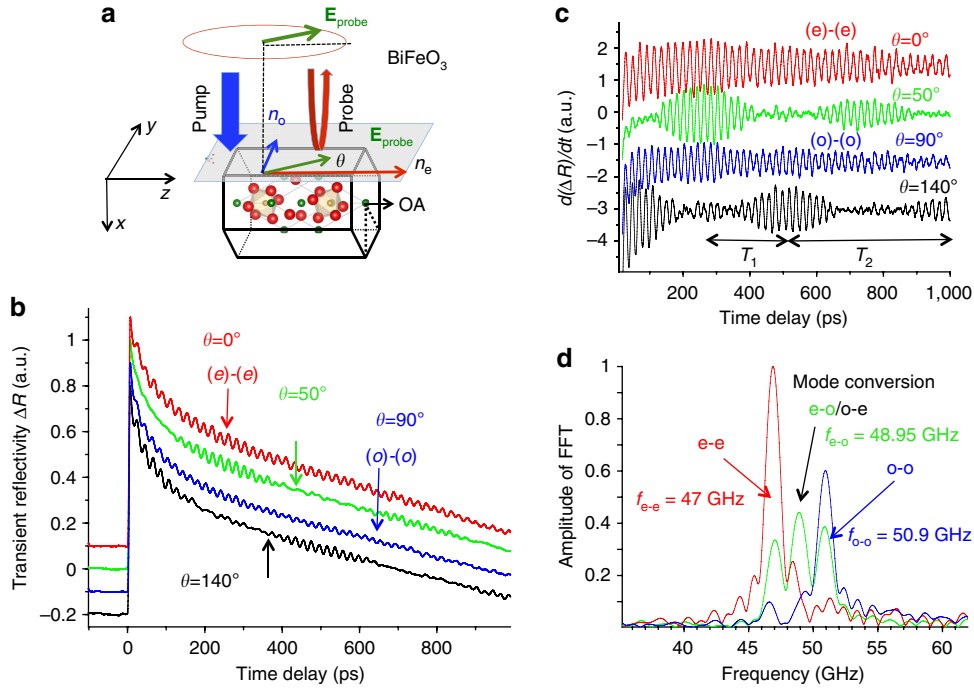

**Figure 2 | Ultrafast acousto-optic mode conversion in BFO.** (**a**) sketch of the pump–probe experiment carried out on BFO with OA along the $z$ axis. The time-resolved Brillouin process is investigated with the probe beam with a variable polarization ($\mathbf{E}_{probe}$). (**b**) Typical transient reflectivity signal with variable linear probe polarization in ferroelectric (BFO). (**c**) Time derivatives of the transient optical signals. $T_1$ and $T_2$ are the beating periods coming from the interferences of the three Brillouin modes (o-o), (e-o)/(o-e) and (e-e). (**d**) FFTs of the signals revealing the (o)-(o) and (e)-(e) processes but also the mode-conversion processes ((e)-(o) and (o)-(e)). The frequencies corresponding to o-o and e-e processes are in full agreement with the expected values and the Brillouin frequency corresponding to the (e)-(o)/(o)-(e) process is exactly centred between the Brillouin frequencies (o)-(o) and (e)-(e) as predicted by the theory.

the pump–probe measurements on the BFO sample are sketched. The pump beam is absorbed within typically 60 nm beneath the surface[33] and set an ultrafast uniaxial pressure that is responsible for the emission of longitudinal acoustic phonons (LA). These propagating acoustic phonons modulate in space and in time the optical indices that the probe beam detects. Figure 2b shows

typical transient optical reflectivity signals as a function of the time delay for various $\theta$ angles of the in-plane light probe polarization ($\mathbf{E}_{probe}$). The first strong peak at a time delay of ≈ 0 ps corresponding to the ultrafast electronic dynamics is followed by long-living oscillations with some beats that are much clearly revealed in the time derivative of the transient signals

(Fig. 2c). A Fast Fourier transform (FFT) allows to extract the frequencies corresponding to the involved Brillouin processes (Fig. 2d). For each light probe propagation with extraordinary (that is, $\theta = 0°$) or ordinary (that is, $\theta = 90°$) refractive index (red and blue curve, respectively, in Fig. 2b,c), only one Brillouin mode is detected and corresponds to the Brillouin frequencies $f_{e-e} = 2n_e V/\lambda = 47$ GHz and $f_{o-o} = 2n_o V/\lambda = 50.9$ GHz, respectively. As expected, the ratio between these two Brillouin frequencies ($f_{e-e}/f_{o-o} = 1.082$) is in excellent agreement with the ratio of refractive indices[34] with $f_{o-o}/f_{e-e} = n_o/n_e \approx 3.05/2.80 = 1.089$ (for a probe wavelength of 590 nm). The observation of both (e)-(e) and (o)-(o) processes is consistent with what has been reported earlier in TeO$_2$ with time-resolved Brillouin investigations[31]. More interestingly, for an arbitrary probe polarization angle (two examples are shown in Fig. 2b–d for $\theta = 50°$ and $\theta = 140°$ corresponding to the green and black curves, respectively), the time-domain Brillouin signals reveal a well-defined central mode in between the (e)-(e) and (o)-(o) modes. A straightforward calculation shows that such extra mode corresponds to the Brillouin mode associated to the light mode conversion with $f_{o-e} = f_{e-o} = (n_o + n_e)V/\lambda = 48.95$ GHz. This simultaneous presence of three Brillouin modes provides in the time domain a complex signal with beating periods corresponding to the difference of Brillouin frequencies $T_1 = 1/(f_{o-o} - f_{e-e}) \sim 256$ ps and $T_2 = 1/(f_{e-o} - f_{e-e}) = 1/(f_{o-o} - f_{e-o}) \sim 512$ ps (see in particular Fig. 2c for the probe angle $\theta = 140°$). To the best of our knowledge, such Brillouin mode-conversion process has never been reported up to now within time-domain Brillouin experiments.

Let us now check whether this mode conversion can be also detected in other uniaxial birefringent media. The textbook birefringent system is the calcite CCO, which has a negative birefringence comparable to that of BFO and a trigonal symmetry such as BFO. Typical transient optical reflectivity signals obtained for CCO covered by a thin chromium film (Fig. 3a) acting as an optoacoustic thermoelastic transducer is shown in Fig. 3b. At the early stage of the signal, the large variation of the transient reflectivity results from the rapid heating of the electrons in the chromium transducer. After a fast electron–phonon thermalization ($\sim 0.5$ ps), a rapid lattice thermal expansion takes place that drives the emission of longitudinal coherent acoustic phonons into the calcite single crystal[26]. These longitudinal acoustic waves propagating in CCO are then detected by the delayed probe pulse giving rise to Brillouin oscillations. For light probe propagation with ordinary or extraordinary refractive index (in blue and red colours, respectively, in Fig. 3b,c), only one Brillouin mode is detected similar to the case of BFO (Fig. 2) or TeO$_2$ (ref. 31) and corresponds to the (o)-(o) and (e)-(e) Brillouin processes, respectively. Both (o)-(o) and (e)-(e) longitudinal acoustic waves clearly revealed by the FFT (Fig. 3d) are found at $f_{o-o} = 2n_o V/\lambda = 41$ GHz and $f_{e-e} = 2n_{eff} V/\lambda = 38.6$ GHz, respectively, where $n_o$ and $n_{eff}$ are the refractive indices for light propagation along the neutral axes (see Fig. 3a) with $n_{eff} = n_o^2 n_e^2/(n_o^2 \sin^2(\beta) + n_e^2 \cos^2(\beta))$[35] where $\beta = 45°$ is the angle between the optical axis (OA) and the surface normal. Using the ordinary and extraordinary refractive indices in Table 1, $n_{eff} = 1.565$ for a probe wavelength of 590 nm, which leads to a ratio of $n_{eff}/n_o = 1.060$ very close to the Brillouin frequencies ratio

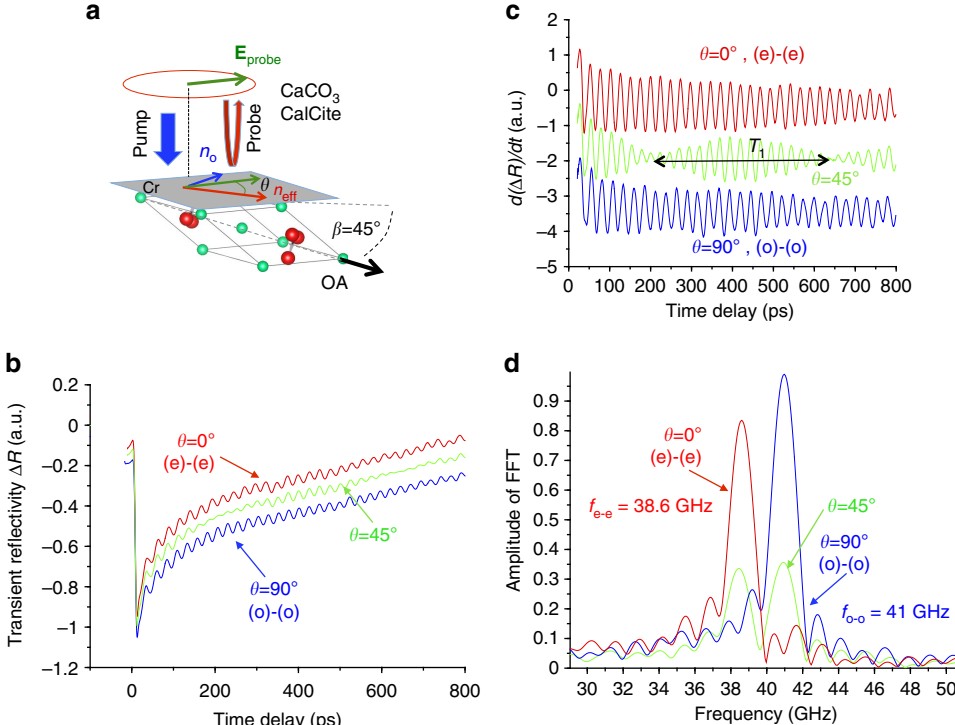

**Figure 3 | Time-resolved Brillouin signal in birefringent calcite crystal.** (**a**) Sketch of the pump–probe experiment carried out on calcite (CCO). The calcium and oxygen atoms are depicted in green and red, respectively, whereas the carbon atoms occupy the centre of the oxygen triangle. The Calcite is covered by a thin chromium film (Cr) which acts as an optoacoustic thermoelastic transducer. The time-resolved Brillouin process is investigated with a controlled variable polarization of the probe beam (**E**$_{probe}$). (**b**) Typical transient reflectivity signal for different linear probe polarizations ($\theta = 90°$ for ordinary and $\theta = 0°$ for extraordinary). (**c**) Time derivatives of the transient optical signals shown in **b**. $T_1$ is the beating period coming from the interferences of the two Brillouin modes (o-o) and (e-e). (**d**) FFTs of the transient optical reflectivity signals revealing the Brillouin frequencies $f_{o-o}$ and $f_{e-e}$ in full agreement with the expected values. For intermediate polarization, both modes are seen simultaneously (green curve).

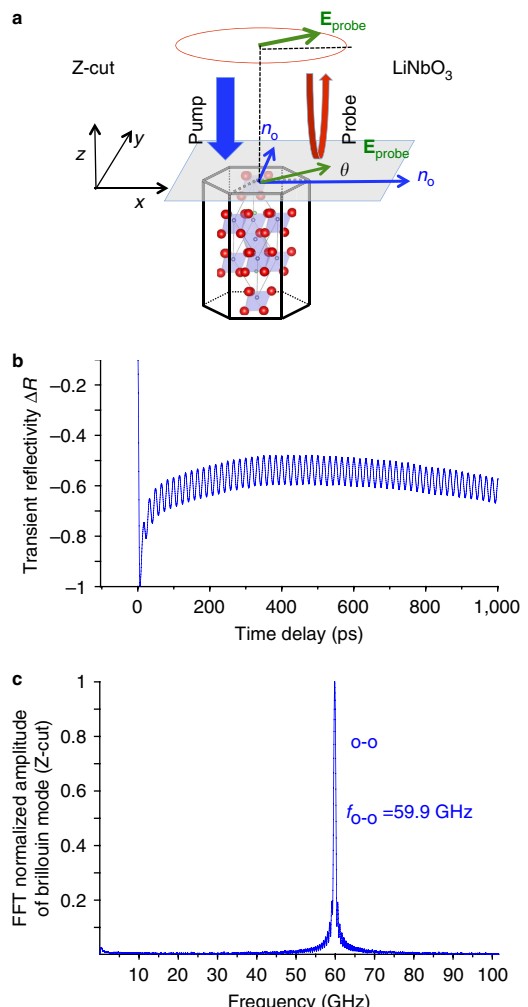

**Figure 4 | Ultrafast elasto-optic detection in Z-cut LNO.** (**a**) Sketch of the pump–probe experiment carried out on Z-cut LNO with OA along the z axis. (**b**) Time-resolved Brillouin signal and (**c**) its corresponding FFT of Z-cut LNO crystal revealing only the (o)-(o) process whatever the probe polarization angle ($E_{probe}$).

acousto-optic effect in LNO, which has a negative birefringence and a trigonal symmetry similar to BFO and CCO but is ferroelectric in contrast to CCO.

The time-resolved Brillouin signal for the three purchased LNO cut directions are given in Fig. 4a,b for Z-cut and in Fig. 5a–d for X- and Y-cut. As expected for the Z-cut isotropic direction, only a single Brillouin mode is observed at $f_B = f_{o-o} = 56.9$ GHz (Fig. 4c). When probing the X-cut anisotropic direction (probe wavelength fixed at 400 nm, to reveal with a larger accuracy the Brillouin splitting), the Brillouin spectrum drastically changes and we evidence three Brillouin peaks (Fig. 5c,e) similar to BFO observations. A straightforward analysis shows that the modes obtained at 74, 75.8 and 77.6 GHz are the Brillouin frequencies corresponding to the (e)-(e), (e)-(o)/(o)-(e) and (o)-(o) processes, respectively. A calculation shows that the central mode at 75.8 GHz perfectly lies in the middle between the two other Brillouin modes with $f_{o-e} = f_{e-o} = (n_o + n_e)V/\lambda$, as it has been predicted (equation (3)). The green curve (Fig. 5c) is obtained for probe angle polarization of $\theta = 120°$ corresponding to the largest mode conversion process signal. When rotating the polarization, it is possible to enhance or diminish the Brillouin signals (that is, cross-section) with clear $\pi$ and $\pi/2$ periodicity observed for the normal processes ((e)-(e) and (o)-(o)), and for the mode conversion ((e)-(o)), respectively. This rich probe angle dependence, also observed for BFO, will be analysed in details in the Discussion.

When probing the Y-cut sample (Fig. 5b), we can also evidence the (e)-(e) and (o)-(o) processes with the Brillouin frequencies of 76.6 and 80.3 GHz, respectively (Fig. 5d,f). However, in this specific configuration, whatever the probe polarization is, no mode conversion process ((e)-(o) or (o)-(e)) is observed as shown by the typical Brillouin spectrum obtained (green curve) in Fig. 5f.

To summarize, we demonstrate here that for both ferroelectric samples (BFO and LNO), it is possible to unambiguously evidence the light mode conversion process induced by GHz coherent acoustic phonons, while in the CCO compound only the two ordinary and extraordinary modes are seen. These remarkable features of ultrafast acousto-optic mode conversion have never been reported in a time-domain Brillouin experiments up to our best knowledge. In the next part we discuss and interpret the results and show that the mode conversion is observed because of specific photoelastic coefficients that birefringent ferroelectric materials exhibit for light propagating along particular crystallographic orientations.

## Discussion

Here we discuss the original ultrafast light mode-conversion evidenced in both birefringent LNO and BFO ferroelectrics. Figure 6 describes the normalized amplitude signal of each Brillouin mode as a function of the full in-plane $\theta$ angle of the light probe polarization (the probe angle dependence of Y-cut LNO is shown in Supplementary Fig. 6 in Supplementary Note 3). Interestingly, a similar angle dependence is found for both compounds (in Fig. 6) anticipating in somehow the universality character of these observations. The magnitude of the Brillouin modes corresponding to (o)-(o) and (e)-(e) processes (including those observed in CCO system as described in Supplementary Fig. 5 in Supplementary Note 3) are $\pi$ periodic, whereas for the Brillouin signal of (o)-(e)/(e)-(o) mode conversion process it is $\pi/2$ periodic. Let us now explain the aforementioned observations and behaviours. To determine the Brillouin processes in a given anisotropic medium, we applied classical electrodynamics to calculate the light intensity modulation ($dI$) collected by the photodetector. We found that for the interpretation of our experimental observations reported here it is not necessary to

$f_{e-e}/f_{o-o} = 1.062$ measured experimentally (Fig. 3d). When the probe polarization is now taken arbitrary (see green curve, $\theta = 45°$ in Fig. 3b–d), no mode conversion (that is, (e)-(o) and (e)-(o)) in between the two (e)-(e) and (o)-(o) modes can be revealed by the FFT in Fig. 3d in contrast to BFO. Indeed, at $f_{o-e} = f_{e-o} = (n_o + n_{eff})V/\lambda = 39.8$ GHz, no signal can be evidenced in Fig. 3d and the amplitude of the FFT rather shows a node. The striking difference between CCO and BFO is also directly seen in the time domain. For a probe polarization at 45°, while two beatings of the transient optical reflectivity signal in BFO are observed, just a single beating of the signal is visible with a period of $T_1 = 1/(f_{o-o} - f_{e-e}) \sim 420$ ps for CCO (Fig. 3c). Additional measurements carried out with variable $\theta$ unambiguously indicate that only the (o)-(o) and (e)-(e) Brillouin modes do exist with their magnitude varying as squared sine and cosine functions of $\theta$, respectively (see Supplementary Fig. 5 in Supplementary Note 3), similar to BFO and LNO, as we will see later on. It is intriguing that in contrast to BFO, CCO shows no evidence of mode conversion. The only main difference between both systems is that BFO is ferroelectric, whereas CCO is not. Knowing that ferroelectrics are very sensitive to strain (here induced by light pulses), let us examine the

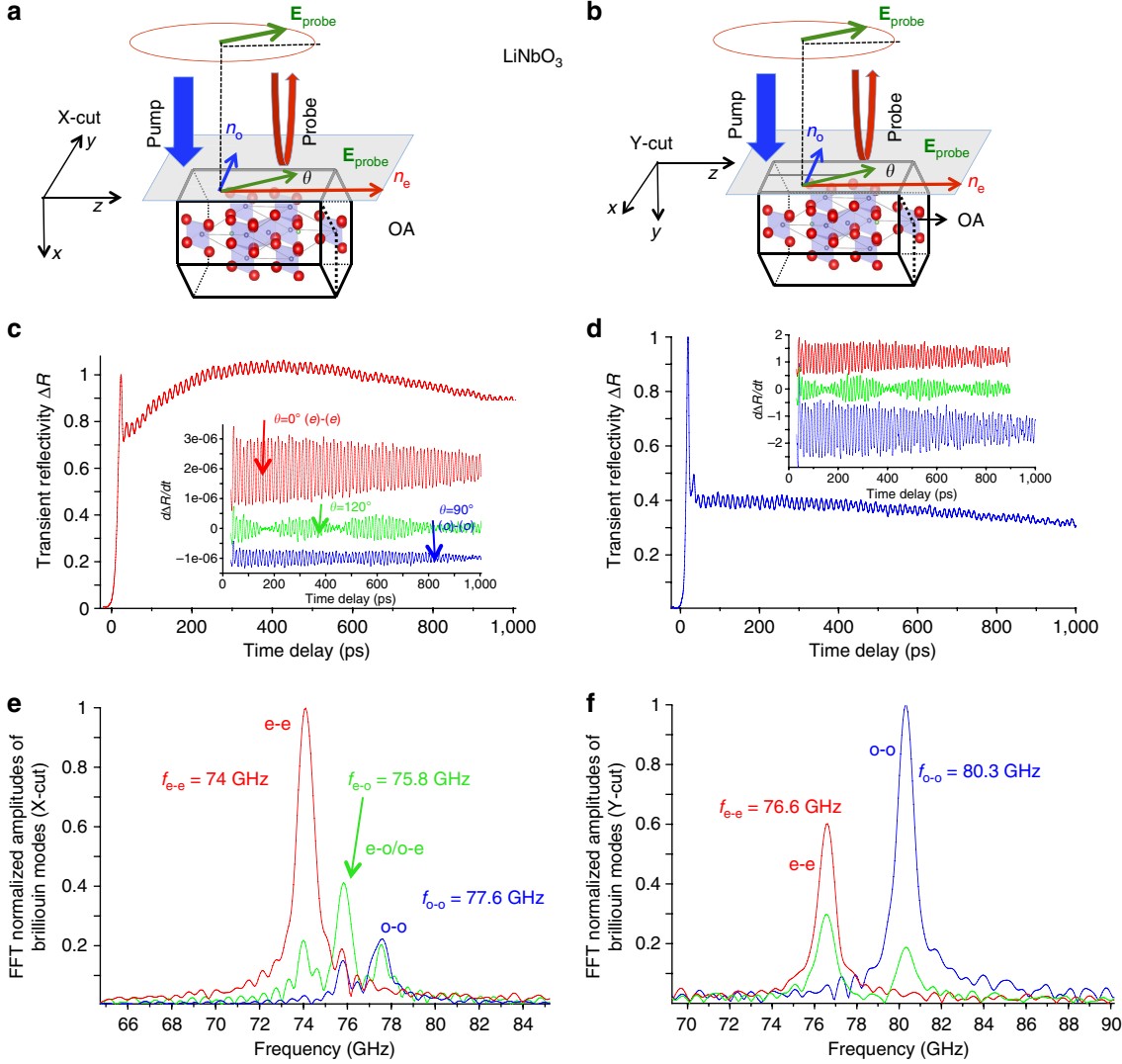

**Figure 5 | Ultrafast elasto-optic mode conversion in X- and Y-cut LNO.** Sketch of the pump–probe experiment carried out on X- (**a**) and Y-cut (**b**) LNO crystal. (**c**,**d**) Corresponding time-resolved Brillouin signals with variable probe polarization angle $\theta$. The red and blue curves are signals obtained for a probe angle $\theta = 0°$ and $\theta = 90°$, respectively, whereas the green curve corresponds to a signal obtained for the probe angle $\theta = 120°$. The insets show the time derivative of the time-resolved Brillouin signals. (**e**) FFT of signals shown in inset of **c**. The FFT of the time-derivative signal of X-cut LNO ($\theta = 120°$) reveals in particular the extraordinary/ordinary ultrafast elasto-optic mode conversion ((e)-(o)/(o)-(e)) together with the normal processes ((o)-(o)) and (e)-(e)). The FFT of Y-cut LNO time-resolved Brillouin signals are given in **f**. In agreement with the theory, whatever the probe electric field angle $\theta$, the FFT demonstrates the absence of the ultrafast elasto-optic mode conversion ((e)-(o)/(o)-(e)) in the Y-cut crystal, whereas the normal processes ((o)-(o)) and (e)-(e)) still exist.

apply advanced theory of photo-elasticity, which accounts for the fact that in general light interacts not with the strain field but with the field of mechanical displacement gradient[22]. Within this consideration, and by definition, the intensity of the scattered probe light is then proportional to the square modulus of the light electric field **E**, which is composed of a reflected electric field ($\mathbf{E}_r = r\mathbf{E}_i$ for beam 1 in Fig. 1d) and a scattered one ($d\mathbf{E}_s$ for beam 2 in Fig. 1d) such as $I \sim (r\mathbf{E}_i + d\mathbf{E}_s)^2$. $r$ is the optical reflection coefficient, which depends on the nature of the light polarization of the incident electric field ($\mathbf{E}_i$), as we will precise later on. The so-called Brillouin signal corresponds to the intensity variations caused by light scattering from the acoustic phonons and is therefore proportional to $dI \sim r\mathbf{E}_i.d\mathbf{E}_s$. In our experiments, the modulus of $d\mathbf{E}_s$ is much smaller than the modulus of $r\mathbf{E}_i$ and, because of this, the contribution to the signal proportional to $d\mathbf{E}_s$ squared is negligible. The expression of the scattered intensity can be generally established by taking into account the photoelastic

tensor $p_{ij}$ of the point symmetry group $3m$ of LNO and BFO with the given axes coordinates ($x,y,z$) described in the inset of Fig. 2a. In the following, the full expressions are established first for a X-cut geometry. In the calculation, we have considered that the incident light probe electric field is $\mathbf{E}_i = E_i \sin(\theta)\mathbf{y} + E_i \cos(\theta)\mathbf{z}$. As a result, the modulation of the intensity caused for instance by the (o)-(o) process is obtained in normal incidence as follow:

$$\begin{aligned} dI_{o-o} &\propto r_o E_y[(1+r_o)p_{21}(1-r_o)E_y] \\ &= r_o(1-r_o^2)p_{21}\sin^2(\theta)E_i^2 \end{aligned} \tag{5}$$

where $r_o E_y$ and $[(1+r_o)p_{21}(1-r_o)E_y]$ are the magnitudes of the reflected (beam 1 in Fig. 1d) and scattered (beam 2 in Fig. 1d) probe light electric field, respectively. In a similar way, the Brillouin intensities for the three other processes can be derived

as:

$$dI_{e-e} \propto r_e(1 - r_e^2)p_{31} \, cos^2(\theta)E_i^2 \qquad (6)$$

$$dI_{e-o} \propto r_o(1 + r_e)p_{41}(1 - r_o) \, sin(2\theta)E_i^2 \qquad (7)$$

$$dI_{o-e} \propto r_e(1 + r_o)p_{41}(1 - r_e) \, sin(2\theta)E_i^2 \qquad (8)$$

where $r_o = (1 - n_o)/(1 + n_o)$ and $r_e = (1 - n_e)/(1 + n_e)$ are the optical reflectivity coefficients for ordinary and extraordinary light. It is now straightforward to see that the (o)-(o) and (e)-(e) processes are indeed $\pi$ periodic as mentioned previously and observed for the X-cut of LNO (Fig. 6b). The above equations also show that the (o)-(e)/(e)-(o) processes have algebraic amplitude that is $\pi$ periodic, whereas the modulus is $\pi/2$ periodic (as observed in Fig. 6), as only a change of sign appears in equations (7) and (8) for a rotation of $\pi/2$, because $sin(2(\theta + \pi/2)) = -sin(2\theta)$. The similar behaviours observed for X-cut LNO and BFO provide us a strong indication that BFO, although not known precisely as mentioned previously (see Methods), has an orientation very close to a X-cut. As an additional support, one can see that the amplitude of the (o)-(e)/(e)-(o) signal obtained for an angle $\theta = 50°$ (green curve in Fig. 2b) is indeed in anti-phase with that obtained for an angle $\theta = 140°$ in very good concordance with theory.

Interestingly, the observation of the mode-converted signals is only possible with pretty large non-diagonal photoelastic coefficient (namely $p_{41}$ in our geometry for the $3m$ point group). If we consider the case of Y-cut crystal, in that geometry, the relevant photoelastic coefficient driving the mode-conversion is now the $p_{51}$, which is zero for symmetry reasons. Consequently, the ultrafast mode conversion process does not exist as clearly reported for the Y-cut of LNO. A complete description was also developed for CCO cut at 45° (see Supplementary Note 4), showing that mode conversion should be observed. However, in that geometry the relevant photoelastic coefficient driving the mode conversion is $p'_{41} = \frac{p_{41} - p_{14}}{2\sqrt{2}} = -0.008$, which is much smaller than that found for LNO (at least 0.155). Even in the case of a CCO crystal with OA axis in plane (cut at 0°, that is, X-cut geometry), the mode conversion would be still unefficient, as $p_{41} = -0.036$ remains a small value.

Furthermore, it is worth noting that the theoretical framework we described previously can reproduce in good agreement the measured time-resolved Brillouin amplitudes. Indeed, as observed in Fig. 6b, the (e)-(e) process ($A_{e-e}^{max}$ as the relative contribution of the (e)-(e) process among the three possible processes, see Table 1) in LNO is roughly three times more efficient that

the (o)-(o) one ($A_{o-o}^{max}$), which is in very good agreement with the ratio of photoelastic coefficients $p_{31}/p_{21} = 0.178/0.072 = 2.4$ as described by the theoretical model (equations (5) and (6)). It is noteworthy that the relative contribution of the maximum peak-to-peak amplitude ($A_{i-i}^{max}$) of the three Brillouin intensities is given in Table 1. This observation indicates that the variation of the acousto-optic process is therefore mainly driven by the photo-elastic coefficients and not by the Fresnel optical reflectivity term ($r_e, r_o$). This is also the case for BFO where only a slight difference exists between $r_e = 0.22$ and $r_o = 0.20$ (at 590 nm). The large $p_{41}$ value of LNO also shows that the mode conversion process is expected to be more efficient than the (o)-(o) process consistently with the experimental measurements ($p_{41}/p_{21} = 0.155/0.072 = 2.1$). As already stressed, the orientation of the studied BFO is not perfectly known inhibiting to determine the $p_{ij}$ coefficients. However, due to the similar behaviour observed in Fig. 6, we can consider that the BFO crystal is close to a X-cut geometry. We have also reported the relative Brillouin amplitude of each of the three processes in Table 1, revealing the importance of $A_{e-o}^{max}$ in both LNO and BFO.

Considering the use of LNO in wide range of photonic applications[9,10] and its commercial availability, our findings open new perspectives for the acousto-optic devices in the GHz ranges. Moreover, it suggests BFO as a promising ferroelectric-based acousto-optic devices with additional potential applications given its lower band-gap compared with that of LNO. Having demonstrated the possibility of inducing light polarization conversion with GHz coherent acoustic phonons generated by a femtosecond laser, we believe that our experiments validate the idea of the GHz–THz acousto-optic devices (illustrated in Fig. 1b) that is likely to be a viable and promising concept. The potential emergence of such devices can take advantage of the fact that LNO compound is already massively used in optics communities. Moreover, it is worth noting that as the acousto-optic mode conversion occurs in the region of the largest strain gradient (corresponding to the largest optical discontinuities[36,37]), the acousto-optic process takes place typically within few nanometres in the case of the femtosecond-induced coherent acoustic phonon pulse discussed here. For LNO, the spatial extension of the coherent acoustic phonons wave packet in the direction perpendicular to the surface is directly controlled by the thermoelastic transducer thickness (Cr layer). In our case, the thickness is 22 nm (see Supplementary Fig. 1), which means that the photoinduced strain is naturally confined only within this absorbing material (LNO is nearly transparent at the pump radiation of 400 nm). For BFO, the spatial extension of the

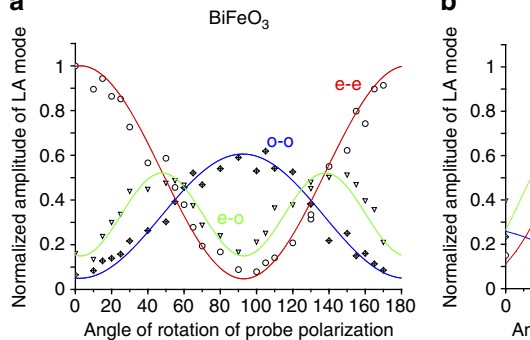
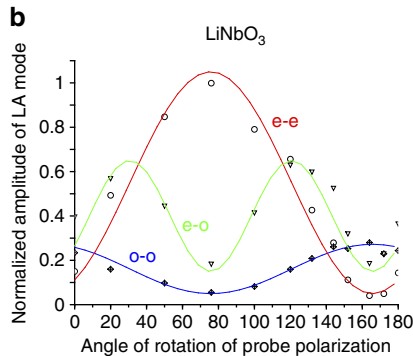

**Figure 6 | Probe light electric field angle dependence of the Brillouin spectrum.** Angle dependence ($\theta$) of the FFT amplitudes of the detected Brillouin modes corresponding to the processes (o)-(o), (e)-(e) and mode conversion processes (o)-(e)/(e)-(o) for (**a**) BFO and (**b**) LNO. The maximum amplitude is normalized to 1. The peak-to-peak amplitudes of each of these normalized curves are named $A_{o-o}^{max}$, $A_{e-e}^{max}$ and $A_{e-o}^{max}$ and their relative values are reported in Table 1. The curves in full lines (red, blue and green) are fitted curves following squared cosinus or sinus functions in agreement with equations (5)–(8), see text.

coherent acoustic phonons wave packet is confined in a region controlled by the optical absorption depth. For a pump energy of 3.1 eV, it is known that the in-depth extension is around 60–80 nm, that is, nearly twice the optical penetration depth of the pump light[38–40]. The in-depth nanometric extension of photoinduced coherent acoustic pulse has been well described in the literature[41]. Based on this theroy, the emitted acoustic pulse with the bipolar shape (BFO, Cr/LNO) sketched in Fig. 1b is fully described in Supplementary Note 2 and Supplementary Fig. 7. Consequently, this acousto-optic mode conversion at the nanoscale can be relevant for future miniaturized devices. Furthermore, giving the time of motion of coherent acoustic waves within this nanometre space scale, the polarization conversion can be controlled at the picosecond time scale. Therefore, by engineering the acoustic pulse front, tuning the polarization state of the scattered light at the picosecond time scale can be envisioned with attractive perspectives of light control through the acousto-optic process. Of course, further efforts in that direction should be done as for instance the current efficiency of the process we revealed remains limited, as the system under investigation is not an optimized device. The light modulation remains small in amplitude ($10^{-4}$), because the Brillouin cross-section is small (the scattered light intensity is much smaller than reflected). Nevertheless, the results we obtained are very promising as when the probe polarization is for example in the region of 45° between the ordinary and extraordinary axes, the light mode conversion Brillouin peak ((e)-(o)/(o)-(e)) reaches similar amplitude, as the (o)-(o) and (e)-(e) processes as shown in Fig. 6.

To conclude, our results reveal for the first time, to our best knowledge, that it is possible to induce a conversion of light polarization with GHz coherent acoustic phonons in birefringent ferroelectrics such as LNO and BFO. Full experimental investigations have been conducted on various crystallographic orientations to clearly evidence the role of optical anisotropy. Our observations are fully supported by the theoretical framework of acousto-optic processes. These results could pave the way of next-generation GHz acousto-optic polarization converter devices with attractive perspectives at the nanoscale. As a direct consequence, we show that an all-optical device for ultrafast manipulation of the light information can be developed by using short laser pulse for launching in ferroelectric media coherent GHz acoustic phonons capable to modify light polarization. Such a device can benefit from the wide knowledge developed on LNO material for its optical properties and be rapidly tested for practical applications. As further direction, we believe that exploring ultrafast acousto-optic effect in ferroelectrics with GHz surface acoustic waves (SAWs) would certainly offer new applications, although current limitations for the generation (detection) of very high frequency SAWs exist, such as the lateral optical diffraction limit for focusing the pump (probe) laser beams. This needs to be overcome to reach the high frequency regime and some alternatives based on surface nanostructurations have been recently reported[42,43].

## Methods

**Sample preparation.** All the samples investigated namely CCO, LNO and BFO are anisotropic uniaxial compounds with a trigonal symmetry and a negative natural birefringence (Table 1). Both LNO and BFO are ferroelectrics with the polarization oriented along the threefold axis, which also corresponds to the c axis in hexagonal setting. CCO is the textbook birefringent compound and is not ferroelectric. To generate coherent acoustic phonons in wide band gap (Table 1) birefringent solids such as the calcite CCO and lithium niobate LNO compounds, we made use of classical ultrafast thermoelastic transducer layer deposited on the surface of CCO and LNO. Our near-ultraviolet laser pump has a maximum energy of 3.1 eV, which is below band gap of these two compounds. We found that for the purposes of our experiments, making use of thermoelastic opto-acoustic conversion is somehow simpler and more efficient than two to use for the coherent

phonon generation electrostriction or inverse piezoelectric effect[29,30], which could be induced even in the transparency region of the tested crystals. It is noteworthy that inverse piezoelectrical effect at GHz frequencies could be initiated by the electric field created by the demodulation/rectification of the pump laser pulse electric field at optical nonlinearity. The selected thermoelastic GHz acoustic phonon emitter is a high-quality thin chromium film (20 nm) deposited by physical vapour deposition (see Supplementary Figs 1 and 2, and Supplementary Note 1). This layer is thin enough to let the probe laser beam penetrate in the transparent CCO or LNO crystals, to detect coherent acoustic phonons that propagate from the free surface towards the inner of the sample. In contrast, for the small band gap birefringent ferroelectric bismuth ferrite BFO (2.6 eV[44]), there is no need of any metallic transducer to generate coherent acoustic phonons, as near-ultraviolet pump energy efficiently generates coherent acoustic phonons[33,45,46]. CCO sample is a single crystal cut with the OA having an angle of around 45° relative to the normal of the surface. This cut corresponds to the naturally cleaved surface allowing to have optical quality for the experiments. Three LNO crystals with typical orientations X-, Y- and Z-cut were purchased (see Supplementary Note 1) and characterized (see Supplementary Fig. 3). We remind that LNO is actually one of the most used optical materials for its acousto-optic, elasto-optic or nonlinear optical properties and is widely used in telecommunication devices. As a matter of fact, different cuts with optical quality are available. Although X- and Y-cut correspond to anisotropic directions with the main OA being perpendicular to the observation direction, the Z-cut is a non-birefringent (isotropic) direction. Because of lack of high-quality and large-enough single BFO crystal, we have performed experiments with polycrystalline BFO sample. As shown recently[33], it is possible to select a specific grain having its OA lying in the surface plane. This geometry is the best one to evidence the optical anisotropy properties of BFO[34]. In contrast to LNO, it is not possible here to distinguish between a pure X-cut or Y-cut, or any other intermediate situation between both cuts.

**Time-resolved optical measurements.** The time-resolved Brillouin scattering experiments were conducted with a two-colour pump–probe scheme following the established procedure[27]. The pump–probe scheme is given in Supplementary Fig. 4 and the details of the experimental procedure are given in Supplementary Note 2. As explained earlier, the coherent acoustic phonons are directly generated in BFO surface, thanks to an above band gap optical pump absorption. For LNO and CCO, a thin chromium film is deposited on the surface to act as an optoacoustic thermoelastic transducer. The laser-induced phonons propagating within the samples and their interaction with the probe light beam are monitored for all configurations (CCO, LNO and BFO) with a below optical band gap probe energy. We have performed experiments in a wide range of probe wavelength (400–800 nm for LNO and CCO, and 560–800 nm for BFO) for which probe light can deeply penetrate in the solid. The pump and probe beams are both in normal incidence ($\psi = \alpha = 0$ in Fig. 1c,d). The probe polarization is accurately controlled with a proper half-wave plate (see Fig. 2a and the details of the optical setup in Supplementary Fig. 4). In the three birefringent solids, only the interaction between longitudinal coherent acoustic phonons (LA) and photons is investigated. Some optical properties including specific photoelastic coefficients for these three solids are given in Table 1.

**Data availability.** All data supporting the findings of this study are available from the authors.

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

## Acknowledgements

Research at Université du Maine was supported by French Ministry of Education and Research, the CNRS and the Region Pays de la Loire (Le Mans Acoustique Program, Ferro-Transducer Project) and Le Mans City (CPER 2007–2013 contract). M.L. thanks the Maine University and the Doctoral School 3MPL for his doctoral grant. G.N., M.G. and J.K. acknowledge support from the Fonds National de Recherche Luxembourg through a PEARL grant (FNR/P12/4853155/Kreisel). I.C.I., B.D. and P.R. acknowledge the funding support through the PHOTOFEnergy project of the Cellule Energie-CNRS. B.D. aknowledges a public grant overseen by the French National Research Agency (ANR) as part of the "Investissements d'Avenir" program (Reference: ANR-10-LABX-0035, Labex NanoSaclay). We thank B. Rufflé and V. Temnov for stimulating discussions.

## Author contributions

M.L. did the time-resolved optical experiments. G.V., I.C and P.R. provided help during the time-resolved optical experiments. I.C.I., M.L., M.E., G.N. and M.G. prepared the samples and realized the necessary characterizations. M.L., G.V., V.E.G., B.D. and P.R. analysed the data. I.C.I., T.P. and J.K. participated in the discussion on the analysis. P.R., V.E.G. and B.D. conceived the project and wrote the paper.
