## [Peer review file · Nature Communications]

Reviewers' Comments:

Reviewer #1 (Remarks to the Author)

Lejman and coworkers reported photo-induced strain in a single grain of polycrystalline sample of BiFeO₃ two years ago. It is expected that the shear strain should modify the optical birefringence. Here the same group succeeds in showing such a modulation of birefringence in BiFeO₃ and LiNbO₃.

The experimental data are of high quality and their analyses are well supported by a theory. This study shows that a pico-second control of light polarization is feasible by using a uniaxial ferroelectric material. I think that this work should be published in Nature Communications, after considering the following point.

I am not sure whether the acousto-optic mode conversion really takes place within a few nanometers. The size of the wave packet of 100-fs laser is estimated to be in the order of ten micrometers. I also wonder whether this estimation is consistent with the theoretical treatment on a polarization control by using the refractive indices for a bulk state.

Reviewer #2 (Remarks to the Author)

The authors report the conversion of fs laser pulses to GHz via optic-mode/acoustic-mode coupling in birefringent ferroelectrics.

I do not think this is of interest to a broad range of readers and hence therefore not to Nature Communications.

It is a very strange mixture of unusually pedagogic text material, such as explaining what Mandelstam-Bruillouin scattering is (though this term is used only in the former Soviet Union), combined with figures, data and equations suitable only for a specialised audience.

The references could be better chosen: For example, there is a related literature by K. A. Nelson et al. (MIT) on impulse optics of pulsed laser sources on ferroelectrics. And of course there is a vast literature on GHz spectroscopy of ferroelectrics from Prague, Rostov, and Moscow.

Finally, regarding the title: I don't know any examples of a NON-BIREFRINGENT ferroelectric. I think that is an impossible oxymoron.

Reviewer #3 (Remarks to the Author)

The authors report on ultrafast acousto-optic mode conversion in birefringent ferroelectrics. The argument is very interesting and can be of interest to a wide public.

The experimental methodology is appropriate and scientifically sound. The theoretical part could be improved in my opinion (see point below).

The paper is well written despite few repetition of the same concepts in the different sections for which I suggest the authors to review the manuscript. In my opinion, in this work there are some points that should be stated and made clearer for it to be published.

1)the authors seem to claim that the effect they present is visible only in ferroelectric materials. They say at the end of the theory part:"In the following, we will show in contrast that the light mode-conversion ((o)-(e)) assisted by GHz coherent acoustic phonons becomes possible in anisotropic ferroelectric materials like LiNbO₃ (LNO)

compound and BiFeO₃

(BFO) system while this process remains not efficient enough in the canonical birefringent CaCO₃ (CCO) crystal."

The sentence is not very clear to me. The phenomenon, as they point out in the SI, depends mainly on the p41 photoelastic coefficient rather than the ferroelectric behavior. They should clarify this in the text and comment more on it.

2) The presence of the Cr thin film affects the system and the phenomenon they are observing is a surface one. The authors provide sound evidence on the quality of the Cr film and I agree that its presence is not interfering with the measurement of the phenomenon. Nevertheless, the paper and explanation would greatly benefit from a full acoustic analysis of a model (either time dependent or harmonic) including with and/or without the thin film. A finite difference or finite element model of the wave behavior would greatly improve the understanding also possibly showing graphically the acoustic wavefronts in the anisotropic crystals making also clear how deep the acoustic signal goes in the crystal for such phenomenon and clarifying the types of acoustic modes involved to a more general public.

3) I suggest to add, at least in the SI, the details on how the authors took the points that they processed with the FFT to extract the wave spectrum. Harmonic inversion could also be helpful to apply in such a signal processing, but FFT is also fine.

For the sake of clarity, the authors should comment more on the beating that is present in fig 3c for 45 deg and fig 4c and 4e.

4) A very good point and interesting measurement would be if a spatial map is taken with the probe beam around the pump (pump fixed, scanning laterally probe or viceversa) to see how far such acoustic phonons can propagate on the surface. That would give an important insight to the phenomenon. This point might require some complication experimentally, so it's only a suggestion and not necessary for the paper although it could improve its scientific strength.

Answer to the referees' comments :

Reviewers' Comments:

Reviewer #1 (Remarks to the Author):

Lejman and coworkers reported photo-induced strain in a single grain of polycrystalline sample of BiFeO₃ two years ago. It is expected that the shear strain should modify the optical birefringence. Here the same group succeeds in showing such a modulation of birefringence in BiFeO₃ and LiNbO₃.

The experimental data are of high quality and their analyses are well supported by a theory. This study shows that a pico-second control of light polarization is feasible by using a uniaxial ferroelectric material. I think that this work should be published in Nature Communications, after considering the following point.

I am not sure whether the acousto-optic mode conversion really takes place within a few nanometers. The size of the wave packet of 100-fs laser is estimated to be in the order of ten micrometers. I also wonder whether this estimation is consistent with the theoretical treatment on a polarization control by using the refractive indices for a bulk state.

Response

We thank Reviewer #1 for his/her very positive comments recommending publication of our manuscript after clarifying his/her concerns. Below is a detailed reply to the questions raised by the Reviewer.

Concerning the spatial extension over which the acousto-optic mode-conversion occurs we can confidently say that it takes place within few nanometers. This is an interesting comment. Actually the acousto-optic interaction/scattering does not take place either in the regions of homogeneous material or in the regions of the homogeneously strained material. Rather, the optical waves are scattered by the gradients of strain. Quantitatively, although in the classical theoretical formulas for the time-domain Brillouin scattering (picosecond ultrasonic interferometry [26,27], the technique which we use in our experiments, the signals are proportional to the spatial overlap integrals of electromagnetic field with acoustic strain, their integration by parts demonstrates straightforward that the interaction is localized at the strain gradients [46]. When the coherent acoustic pulses are photoexcited by light absorption near the mechanically free surface they commonly have two dominating consecutive phase of opposite polarity [29]. The duration of the transition front between two phases of the coherent acoustic pulse launched nearly ideally flat mechanically free surface grows with increasing duration of the pump laser pulses. In the case of the femtosecond pump laser pulses this duration is short and one could need to account for the possible roughness of the material surface. For example, even the roughness of 1 nm broadens the duration of the transition front in the coherent acoustic pulse to the duration of sound propagation across 2 nm depth,

which exceeds the duration of pump laser pulses in our experiments. The duration of the transition front between the acoustic phases of opposite polarity could be potentially broadened by the acoustic wave attenuation. However there are no signs of any significant attenuation of acoustic waves at the spatial scale of several micrometers in our experiments (see Figs. 2 – 4). Thus in our experiments the spatial length of the transition front is expected to be of few nanometers and is much shorter than the durations of the leading and trailing acoustic phases surrounding it, which are controlled by the penetration depth of light in BFO (about 30-40 nm [48-50]) or by the thickness of the metallic optoacoustic transducer deposited on LNO (22 nm, see Supplementary Figure 2). Thus the acoustic strain gradients in the transition front exceed at least by an order of magnitude the strain gradients in the leading and trailing fronts of the bipolar acoustic pulse (see qualitative presentation of pulse profile in Fig. 3 (b) and the calculated profile given in Supplementary Figure 7) and the mode conversion of the light pulses in our experiments is expected to take place at the spatial scale of few nanometers. This estimation is consistent with the theoretical treatment on a polarization control by using the refractive indices for a bulk state. It is worth noting that the recent experiments [47] have confirmed that the Brillouin scattering by the sharp front of the coherent acoustic pulse dominates over the light scattering by its other parts. This information on the spatial scale of the light mode-conversion is important regarding possible future development of miniaturized acousto-optic devices. We have added in page 10 additional references [46-47] confirming that the mode-conversion takes place in the domain of highest acoustic strain gradient.

We have also added in page 10 a new sentence in the discussion with the proper new references concerning the pump optical absorption in BFO as well as the description of the modeling of the photoinduced strain [48-51].

“For LNO, the spatial extension of the coherent acoustic phonons wave packet in the direction perpendicular to the surface is controlled by the thermos-elastic transducer thickness (Cr layer). In our case the thickness is 22 nm (see Supplementary Figure 2) which means that the photo-induced strain is naturally confined only within this absorbing material (LNO is nearly transparent at the pump radiation of 400nm). For BFO, the spatial extension of the coherent acoustic phonons wave packet is confined in a region controlled by the optical absorption depth. For a pump energy of 3.1 eV, it is known that the in-depth extension is around 60-80 nm, i.e. nearly twice the optical penetration depth of the pump light [48-50]. The in-depth nanometric extension of photoinduced coherent acoustic pulse has been well described in the literature [51]. Based on this theory, the emitted acoustic pulse with the bipolar shape (BFO, Cr/LNO) sketched in Fig.1b is fully described in Supplementary Note 3 and Supplementary Figure 7.”

We also added the above discussion in the Supplementary Note 3 since we think it is worth to be given to the reader.

Reviewer #2 (Remarks to the Author):

The authors report the conversion of fs laser pulses to GHz via optic-mode/acoustic-mode coupling in birefringent ferroelectrics.

I do not think this is of interest to a broad range of readers and hence therefore not to Nature Communications.

It is a very strange mixture of unusually pedagogic text material, such as explaining what Mandelshtam-Bruillouin scattering is (though this term is used only in the former Soviet Union), combined with figures, data and equations suitable only for a specialised audience.

The references could be better chosen: For example, there is a related literature by K. A. Nelson et al. (MIT) on impulse optics of pulsed laser sources on ferroelectrics. And of course there is a vast literature on GHz spectroscopy of ferroelectrics from Prague, Rostov, and Moscow.

Finally, regarding the title: I don't know any examples of a NON-BIREFRINGENT ferroelectric. I think that is an impossible oxymoron.

Response

We thank the Reviewer #2 for the reading of our manuscript. However, we respectfully disagree with the statement of Reviewer #2 "*I do not think this is of interest to a broad range of readers and hence therefore not to Nature Communications.*"

Note first that this statement contradicts the following sentences of Reviewer #3 "*The authors report on ultrafast acousto-optic mode conversion in birefringent ferroelectrics. The argument is very interesting and can be of interest to a wide public.*" as well as the comments of Reviewer #1 "*The experimental data are of high quality and their analyses are well supported by a theory. This study shows that a pico-second control of light polarization is feasible by using a uniaxial ferroelectric material. I think that this work should be published in Nature Communications*".

Let us recall why we believe this work is of interest for the readers of Nature Communications. This is the first experimental work showing unambiguously that light polarization can be converted using ultrafast GHz and coherent acoustic waves and this is likely to take place within a few nanometer spatial scale. This experimental evidence is fully supported by theoretical calculations we did and which reveal that specific photoelastic coefficients are involved with those coefficients being much larger in case of ferroelectric and multiferroic materials stressing the interest of these materials for optics and acoustics. We also propose a promising concept for an all-optical ultrafast light modulation device that can be realized/developed quickly using the already available ferroelectric LiNbO₃. Consequently we think that our manuscript provides new and original findings which are of interest for optics, acoustics, materials science, ferroelectrics and multiferroics as well as devices communities.

Besides, we thank the Reviewer #2 for his/her comment on the missing reference that could be added. There is a general huge interest of ultrafast coupling of lattice modes and electromagnetic waves. The Reviewer is right when mentioning the work

of the group of K. Nelson (MIT, Boston). They have in particular developed a comprehensive description of the coherent optical phonons interaction/coupling with ultra-short light pulses and have shown how it was possible to control in the space and time domains the polaritons dynamics (polaritonics). In particular, polaritonics has opened some important directions towards the THz light source. We have therefore added a reference related to polaritonics in the end of the introduction :

Moreover, we also reveal that in addition to usual LNO material, smaller band gap ferroelectrics like the prototypical BFO compound, which displays comparable photoelastic responses, hold promises in ferroelectric-based acousto-optic devices enriching their already wide photoferroelectric [14] as well as polaritonic (new references [15,16]) potentialities.

Furthermore, we do not understand what the Reviewer means when he/she mentions GHz spectroscopy of ferroelectrics. There are of course a lot of studies like IR, microwaves and even THz spectroscopic measurements with ferroelectrics but the physics deals with thermal phonons while our research topic is based on coherent acoustic phonons and their interaction with electromagnetic waves (visible range) in the frame of acousto-optic physics. According to our best knowledge, the reported by us ultrafast light mode-conversion assisted by tens-GHz coherent acoustic phonons has never been observed before, despite the existence of many schemes of acousto-optic modulation already investigated (see References [1-10]) and we believe that our findings is potentially of a reach interest for next generation acousto-optic devices.

Concerning the title, the Reviewer #2 is right all ferroelectrics are birefringent materials. However as we believe that our work is of interest for various communities including materials science but also optics or acoustics, we preferred to specify that ferroelectrics are indeed birefringent in the title.

Reviewer #3 (Remarks to the Author):

The authors report on ultrafast acousto-optic mode conversion in birefringent ferroelectrics. The argument is very interesting and can be of interest to a wide public.

The experimental methodology is appropriate and scientifically sound. The theoretical part could be improved in my opinion (see point below).

The paper is well written despite few repetition of the same concepts in the different sections for which I suggest the authors to review the manuscript. In my opinion, in this work there are some points that should be stated and made clearer for it to be published.

Response

We would like to thank Reviewer #3 for his/her positive and valuable comments recognizing the interest of our findings and their impact to a broad readership. We made in the following a point-by-point and detailed answer to clarify his/her concerns

and addressed them in the revised version.

Comment 1) the authors seem to claim that the effect they present is visible only in ferroelectric materials. They say at the end of the theory part: "In the following, we will show in contrast that the light mode-conversion ((o)-(e)) assisted by GHz coherent acoustic phonons becomes possible in anisotropic ferroelectric materials like LiNbO₃ (LNO) compound and BiFeO₃ (BFO) system while this process remains not efficient enough in the canonical birefringent CaCO₃ (CCO) crystal."

The sentence is not very clear to me. The phenomenon, as they point out in the SI, depends mainly on the p₄₁ photoelastic coefficient rather than the ferroelectric behavior. They should clarify this in the text and comment more on it.

Response

The Reviewer is right. Indeed the mode conversion effect is mainly driven by the peculiar photoelastic coefficient, P₄₁, which, for both ferroelectric materials BFO and LNO we studied, is much larger than the one of CCO, the canonical birefringent material. We believe there is certainly a relationship between these large P₄₁ coefficients and the ferroelectricity however we did not have established it yet. We have therefore modified the text as:

In the following, we will show in contrast that the light mode-conversion ((o)-(e)) assisted by GHz coherent acoustic phonons becomes possible in ferroelectric materials like LiNbO₃ (LNO) and BiFeO₃ (BFO) while this process remains not efficient enough in the canonical birefringent CaCO₃ (CCO) crystal. As we will see, this peculiar behavior comes from exceptionally large P₄₁ coefficients that BFO and LNO ferroelectrics have and which are probably likely linked to the ferroelectric properties although no direct relationship has been established yet.

Comment 2) The presence of the Cr thin film affects the system and the phenomenon they are observing is a surface one. The authors provide sound evidence on the quality of the Cr film and I agree that its presence is not interfering with the measurement of the phenomenon. Nevertheless, the paper and explanation would greatly benefit from a full acoustic analysis of a model (either time dependent or harmonic) including with and/or without the thin film. A finite difference or finite element model of the wave behavior would greatly improve the understanding also possibly showing graphically the acoustic wavefronts in the anisotropic crystals making also clear how deep the acoustic signal goes in the crystal for such phenomenon and clarifying the types of acoustic modes involved to a more general public.

Response

We thank the Reviewer for this comment. Actually, the numerical evaluations are not necessary. Indeed, the analytical theories [26, 27, 51] provide a clear insight in the phenomenon of Brillouin light scattering by coherent acoustic pulses. For the case of the BFO sample with the mechanically free surface, the acoustic pulses are photo-generated inside BFO the most relevant references are [34, 35, 48-50] where all the

parameters (both temporal and spatial) of the acousto-optic interaction in BFO are given (for example the penetration lengths of the pump (400nm) in BFO is 30-40nm [48,49]) which in turns give an in-depth extension of 60-80nm. The detailed analysis of the case of the metallic opto-acoustic transducer can be found in [51]. To address the suggestions of the Reviewer we have introduced additional reference [51] in the text as well as a new sentence with additional details in Supplementary Note 3

“The in-depth nanometric extension of photoinduced coherent acoustic pulse has been well described in the literature [51]. Based on this theory, the emitted acoustic pulse with the bipolar shape (BFO, Cr/LNO) sketched in Fig.1b is fully described in Supplementary Note 3 and Supplementary Figure 7.”

We also added a detailed discussion in the Supplementary Informations (Note 3) a novel Figure (Supplementary Figure 7) as suggested by the Reviewer where the characteristic spatial scales of our experiments in BFO and Cr/LNO are explicitly calculated and represented.

Comment 3) I suggest to add, at least in the SI, the details on how the authors took the points that they processed with the FFT to extract the wave spectrum. Harmonic inversion could also be helpful to apply in such a signal processing, but FFT is also fine.

For the sake of clarity, the authors should comment more on the beating that is present in fig 3c for 45 deg and fig 4c and 4e.

Response

Following the suggestion of the Reviewer we added a sentence concerning the signal processing in the SI in complement to our initial text as following:

As a main information, the quality of our signal (long-living Brillouin oscillation up to more than 1000ps (1ns)) provides us a very good frequency resolution (better than 1 GHz) which permits to unambiguously distinguish all the Brillouin components without any advanced signal processing. In the FFT treatment of Brillouin signals, we applied the traditional zero-padding method for the frequency interpolation.

Concerning the beatings structure in the time domain signals, we added supplemental sentences at different places to clearly show the relation with the FFT signal

At the end of the first paragraph of the part Results (discussion about BFO):

This simultaneous presence of three Brillouin modes provides in the time domain a complex signal with beating periods corresponding to the differences of Brillouin frequencies $T_1=1/(f_{i-o}-f_{e-e}) \sim 256$ ps and $T_2=1/(f_{e-o}-f_{e-e}) \sim 512$ ps (see in particular Fig 2c for the probe angle $\theta=140^\circ$).

In the second paragraph of the part Results (discussion about CCO), we have installed a new sentence discussing the beating.

The striking difference between CCO and BFO is also directly seen in the time domain. For a probe polarization at 45°, while two beatings of the transient optical reflectivity signal in BFO are observed, just a single beating of the signal is visible with a period of $T_1=1/(f_{o-o}-f_{e-e}) \sim 420$ ps for CCO.

Concerning the fig 4c,e on LNO, we have also the presence of beatings coming from the interferences of the different Brillouin components. This is clearly seen for the Y-cut crystal with only one beating period ($T_1=1/(f_{o-o}-f_{e-e}) \sim 270$ ps. For the X-cut, the two periods ($T_1=1/(f_{o-o}-f_{e-e}) \sim 384$ ps and $T_2=1/(f_{e-o}-f_{e-e}) \sim 768$ ps) are not so easy visible in the time domain in LNO than in the case of BFO, simply because of a much smaller magnitudes of the Brillouin components (e-o) compared to those of (e-e) and (o-o) components in LNO.

Comment 4) A very good point and interesting measurement would be if a spatial map is taken with the probe beam around the pump (pump fixed, scanning laterally probe or viceversa) to see how far such acoustic phonons can propagate on the surface. That would give an important insight to the phenomenon. This point might require some complication experimentally, so it's only a suggestion and not necessary for the paper although it could improve its scientific strength.

Response

The Reviewer is right that photoinduced SAW in ferroelectrics would also be relevant for applications. Performing photoexcitation and photodetection of GHz SAW is currently very rare and is the object of intense development. We are currently working in this direction but it is by itself a new research program. Nevertheless, we have included two additional references dealing with recent progress in high frequency SAWs as following:

As further direction, we believe that exploring ultrafast acousto-optic effect with GHz surface acoustic waves (SAWs) would certainly offer new applications although current limitations for the generation (detection) of very high frequency SAWs such as the lateral optical diffraction limit for focusing the pump (probe) laser beams. This needs to be overcome to reach the high frequency regime and some alternatives based on surface nanostructuring have been recently reported [52-53].

Reviewers' Comments:

Reviewer #3 (Remarks to the Author)

In the revised version submitted, the authors have addressed the main points and I think the paper should be accepted for publication.